# Incidence of Rifampicin Resistance in Periprosthetic Joint Infection: A Single-Centre Cohort Study on 238 Patients

**DOI:** 10.3390/antibiotics12101499

**Published:** 2023-09-30

**Authors:** Stergios Lazarinis, Nils P. Hailer, Josef D. Järhult, Anders Brüggemann

**Affiliations:** 1Department of Surgical Sciences, Orthopaedics, Uppsala University, SE-751 85 Uppsala, Sweden; nils.hailer@uu.se (N.P.H.); anders.bruggemann@uu.se (A.B.); 2Department of Medical Sciences, Zoonosis Science Center, Uppsala University, SE-751 85 Uppsala, Sweden; josef.jarhult@medsci.uu.se

**Keywords:** incidence, periprosthetic joint infection, rifampicin resistance, treatment failure

## Abstract

Background. Rifampicin is a pillar in the treatment of periprosthetic joint infection (PJI). However, rifampicin resistance is an increasing threat to PJI treatment. This study explores the incidence of rifampicin-resistant bacteria over time in a Swedish tertiary referral centre and the association of rifampicin resistance with infection-free survival after PJI. Methods. The study included 238 staphylococcal PJIs treated between 2001 and 2020 for which susceptibility data for rifampicin were available. Data on causative bacteria, rifampicin resistance, treatment, and outcome were obtained. Kaplan–Meier survival analysis and Cox regression modelling estimated the infection-free cumulative survival and adjusted hazard ratios (HRs) for the risk of treatment failure. Results. Rifampicin-resistant causative bacteria were identified in 40 cases (17%). The proportion of rifampicin-resistant agents decreased from 24% in 2010–2015 to 12% in 2016–2020. The 2-year infection-free survival rates were 78.6% (95% CI, 66.4–93.1%) for the rifampicin-resistant group and 90.0% (95% CI, 85.8–94.4%) for the rifampicin-sensitive group. Patients with PJI caused by rifampicin-resistant bacteria had an increased risk of treatment failure (adjusted HR, 4.2; 95% CI, 1.7–10.3). Conclusions. The incidence of PJI caused by rifampicin-resistant bacteria did not increase over the past 20 years. The risk of treatment failure in PJI caused by rifampicin-resistant bacteria is more than four times that caused by rifampicin-sensitive bacteria, highlighting the importance of limiting the development of rifampicin resistance.

## 1. Introduction

Periprosthetic joint infection (PJI) is a serious, potentially deadly, and the most common early complication after arthroplasty surgery [1,2]. The incidence of PJI seems to have increased in many developed countries including those in the European Union [3], and a recent report form the Nordic countries indicated that the risk of revision due to PJI has doubled within the last decade [4]. Several algorithms to prevent, diagnose, and treat PJI have been proposed [5,6,7,8,9].

The treatment of PJI, regardless of whether it presents early, delayed, or at the chronic stage, requires surgical intervention in combination with antibiotic treatment [10]. Biofilm formation on the surface of implants is an integral part of PJI pathophysiology and poses a challenge to the successful treatment of PJI [11].

One of the most effective antibiotics against biofilm is rifampicin; it has excellent penetration properties and biofilm-degrading activity [12,13,14]. Several studies show satisfactory or good results after the prolonged use of antibiotic combination therapy with rifampicin, especially in PJIs caused by staphylococci [15,16,17].

Antibiotic resistance is one of the most serious threats to human health [18,19]. As bacteria evolve and adapt to the selective pressure of antibacterial agents, our formidable antibiotics arsenal is losing its potency. During the last decades, more and more devices are implanted in humans, and implant-related infections are a serious and dramatic complication after those interventions. In particular, resistance to biofilm-active antibiotics such as rifampicin jeopardises the treatment of implant-related infections and, thus, PJI [20]. However, studies on the trends of the incidence of rifampicin resistance over time and the effect of rifampicin resistance on treatment failure in PJI are scarce.

In this observational study, we aimed to answer the following questions:Has the incidence of PJI caused by rifampicin-resistant bacteria changed over the last 20 years in a single tertiary referral centre in Sweden?Is there an association between the rifampicin resistance of the causative bacteria and treatment failure in PJI?

## 2. Results

### 2.1. Cohort Characteristics

In total, 238 patients with 238 PJIs were included in the data analysis (Table 1); 137 (58%) patients were males, and the mean age of the cohort was 71 years (range, 16–96 years). A total of 154 hips (65%) and 84 knees (35%) were included, and 149 (63%) patients underwent primary arthroplasty before the index surgery. DAIR was the most frequent (67%) index treatment for PJI, followed by two-stage revision (26%) and one-stage revision (7%). Coagulase-negative staphylococci (CoNS) were the most frequently identified organisms (55%).

### 2.2. Incidence of Rifampicin Resistance

Forty PJIs (17%) caused by rifampicin-resistant bacteria were identified. The index operation was two-stage revision in 58% of the cases in the rifampicin-resistant group. DAIR dominated (73%) in the rifampicin-sensitive group. Coagulase-negative staphylococci (CoNS) dominated in the rifampicin-resistant group (93%) (Table 1). We explored our data, divided into 5-year intervals, to identify a possible trend in the proportion of rifampicin-resistant agents over time (Figure 1). An increase in the proportion of rifampicin-resistant bacteria was found the between periods 2001–2005 (13.3%), 2006–2010 (15.5%), and 2011–2015 (24.0%). During the last decade, the proportion of rifampicin-resistant bacteria decreased from 24.0% between 2011 and 2015 to 12.1% between 2016 and 2020. We did not find any statistically significant difference between the chosen time periods. Most of the PJIs caused by rifampicin-resistant bacteria (77.5%) were observed in the last 10 years, i.e., between 2011 and 2020.

### 2.3. Treatment Outcomes

The infection-free survival for the entire cohort (*n* = 238) was 88.7% (*n* = 211). In PJIs caused by rifampicin-resistant bacteria, infection relapse and treatment failure occurred in 20.0% (*n* = 8) of cases compared with 9.6% (*n* = 19) in PJIs caused by rifampicin-sensitive bacteria. The 2-year infection-free survival was estimated at 78.6% (95% CI, 66.4–93.1%) for the rifampicin-resistant group and 90.0% (95% CI, 85.8–94.4%) for the rifampicin-sensitive group (*p* = 0.07, Figure 2).

The adjusted HR for treatment failure was 4.2 (95% CI, 1.7–10.3, *p* = 0.002) for patients with infection caused by rifampicin-resistant bacteria.

Seventy-three percent of the patients in the rifampicin-sensitive group were treated with rifampicin for a median duration of 12 weeks. In total, 128 of 145 patients deemed suitable for treatment with rifampicin fulfilled their treatment as planned, while 17 had to abort their treatment as indicated by a treatment for less than 6 weeks. When analysing the effect of rifampicin treatment in PJIs caused by rifampicin-sensitive bacteria, we found that patients treated with rifampicin had a lower risk of infection relapse than those who did not receive rifampicin, but the results were not statistically significant (HR 0.5, 95% CI: 0.2–1.2, *p* = 0.11).

We also performed a subgroup analysis on PJIs caused only by CoNS (*n* = 131). We found the same trend of a decreasing incidence of resistance to rifampicin during the last decade when we investigated all staphylococci (24.0% in 2011–2015 versus 12.1% in 2016–2020), and when we analysed PJIs caused only by CoNS (36.7% in 2011–2015 versus 20.8% in 2016–2020). Since most rifampicin-resistant bacteria were found in the subgroup of patients where CoNS were the causative bacteria, we performed an analysis only investigating the relapse-free survival in this patient group (Figure 3). The findings from this analysis differed only slightly from those including the whole cohort of PJI. An additional stratified analysis investigating the differences in survival for the different time periods revealed no significant difference, with the exception of the first period (2001–2005), during which there were no failures; hence, the survival including 95% CIs was 100% (Appendix A, Appendix A).

## 3. Discussion

The primary outcome of our study was the incidence of PJI caused by rifampicin-resistant bacteria over the last 20 years at our centre. We found a decreasing trend for the proportion of PJIs caused by rifampicin-resistant bacteria between 2011–2015 and 2016–2020. Our secondary outcome was to investigate the association between the rifampicin resistance of the causative bacteria and treatment results for PJI. We found that the risk of infection relapse was 4.2 times higher for patients with an infection caused by rifampicin-resistant bacteria compared with those who had PJIs caused by rifampicin-sensitive bacteria.

Over the last decades, there has been an alarming increase in antibiotic resistance, and an increase in multidrug-resistant pathogens causing life-threatening human infections has been observed [18,19]. Rifampicin is one of the most important drugs used in the treatment of PJI; thus, resistance to rifampicin is a threat to the management of PJI. Fröschen et al. [21] reported a stable average rifampicin resistance per year of 24.4% in PJIs caused by CoNS in the last 6 years (2016–2021) in Germany. Contrary to these results and the global trend of increasing antibiotic resistance, we found a decreasing trend of PJIs caused by rifampicin-resistant bacteria in the last decade. One explanation for our results could be that Scandinavian countries, especially Sweden, have developed strategies against antibiotic resistance. STRAMA, the Swedish strategic programme against antibiotic resistance, was implemented throughout the health care system in the past decades [22]. In addition, the Swedish Infectious Diseases Association publishes and regularly updates guidelines on the treatment of implant-associated infections, including the specific indications for the use of rifampicin in PJI [23]. The results of our study may be explained by good adherence to STRAMA and the Infectious Diseases Association’s guidelines in the last 5 years in Sweden generally and especially in our region and hospital.

Rifampicin is an orally well-absorbed antibiotic that is metabolised in the liver. Its bactericidal activity is mediated by inhibiting the bacterial RNA polymerase. It exhibits a strong post-antibiotic effect with good results against susceptible bacteria [24]. It is often used in combination with other antibiotics in the treatment of PJI in order to increase the overall effectiveness and to prevent the development of resistance. Understanding the pharmacokinetic and pharmacodynamic properties of rifampicin is crucial for optimising dosing regimens, minimising side effects, and preventing the development of antibiotic resistance. As indicated by the length of rifampicin treatment, most patients in our study fulfilled their treatment. However, some patients were never treated with rifampicin, even though the identified causative bacteria were rifampicin-sensitive. This might be due to some patient-related factors that we cannot account for, such as interactions with other pharmaceuticals.

Rifampicin resistance develops through a single-step mutation in the *rpoB* gene encoding the β-subunit of bacterial DNA-dependent RNA polymerase [25,26,27]. Resistance can emerge rapidly, especially when rifampicin is used as a monotherapy or when the bacterial load is high. Therefore, rifampicin for PJI treatment should always be used in combination with other antibiotics and should not be administrated until a few days after PJI surgery once the bacterial burden has been reduced by surgical revision and initial antibiotic treatment [15,28,29,30]. In recent years, our PJI team has followed national and local guidelines, using rifampicin in combination with other antibiotics and withholding treatment until the bacterial load is low, as judged by the absence of secretion from the surgical wound. Other studies have confirmed the importance of adhering to this strategy in the usage of rifampicin, since the delayed administration of rifampicin significantly reduces the emergence of rifampicin resistance [31].

Reports from South Africa and China have recently shown a dramatic increase in rifampicin- and methicillin-resistant *Staphylococcus aureus* (MRSA) [32,33]. The incidence of MRSA has been comparatively low in Sweden, likely due to strict hygiene rules and the isolation of patients with MRSA infections, as well as prudent antibiotic use. In our study, we found only one patient with a PJI caused by MRSA, and similar underlying factors could also explain the low incidence of rifampicin resistance in our cohort. The low prevalence of MRSA might influence the external validity of our study, since other countries do—as mentioned above—present with a much higher prevalence of MRSA.

Resistance is also a major concern in the treatment of CoNS and varies in its prevalence, even when comparing similar populations [34]. Shabana et al. highlight the possible need for adjustments in treatments when facing PJIs caused by resistant CoNS [35]. One-third of all CoNS included in this study were resistant to rifampicin, which is in line with previous findings [36]. Unfortunately, we lack data on resistance to antibiotics other than rifampicin. For instance, in our clinical practice, most CoNS are methicillin-resistant, but we did not perform an analysis of the prevalence of MRCoNS in this study due to its explicit focus on resistance to rifampicin. Furthermore, the presence of methicillin resistance does not guide the treatment of CoNS in the clinical setting.

The secondary outcome of our study was to investigate the association between rifampicin-resistant causative bacteria and treatment failure in PJI. The literature on the effect of rifampicin resistance on PJI treatment is scarce. A study by Krizsan et al. [37] on 73 patients undergoing two-stage revision for PJI found that rifampicin resistance significantly reduced the probability of treatment success. Our results show a lower treatment success rate in PJIs caused by rifampicin-resistant bacteria compared with those caused by rifampicin-sensitive agents and are thus in line with the limited current literature. However, our results are at risk of bias due to the surgical method applied; most rifampicin-resistant PJIs were treated with two-stage revision surgery, while the majority of patients with a PJI caused by rifampicin-sensitive bacteria were treated with DAIR. In general, two-stage revision surgery shows better treatment outcomes than DAIR, at least when comparing infections caused by the same bacteria and equal patient- and surgery-related risk factors. We are unable to control for all of the underlying factors determining the surgical method, but we included the type of surgery prior to the index surgery as a proxy measure for the cases’ complexity in our Cox regression. However, this does not fully eliminate the bias; hence, there might be an even greater difference in infection-free survival between these two groups than that established in our study.

A multicentre study by Achermann et al. identified several factors associated with the development of rifampicin resistance in staphylococcal PJI such as male sex, ≥3 previous surgical interventions, PJI treatment during high bacterial load, and inadequate rifampicin therapy [38]. In our study, we were unable to find any statistically significant differences between the rifampicin-sensitive and the rifampicin-resistant groups regarding these factors. Nonetheless, we did notice that in the rifampicin-resistant group, two-stage revision dominated as the index surgery compared to the rifampicin-sensitive group where DAIR was the main index surgery. This implies that more previous surgical interventions had been performed in patients with rifampicin-resistant bacteria.

Rifampicin, with its ability to penetrate biofilm, is one of the main antibiotics used in the treatment of PJIs. Amongst the cases with rifampicin-sensitive bacteria, we found that patients treated with rifampicin had a lower risk of treatment failure than those who did not receive rifampicin, as indicated by the 2-year survival rates (92.1%, 95%CI: 87.7–96.7 vs. 82.5%, 95%CI: 72.2–94.3). Our results are in line with those reported by Karlsen et al. [39] in a randomised controlled trial on 48 patients treated with DAIR due to acute staphylococcal PJI, showing that rifampicin conferred a significant advantage when added to standard antibiotic treatment. Our findings support the continued use of rifampicin in the treatment of PJI, confirming the efficacy of this drug against biofilm-induced implant infections [40].

Our study has several limitations. The limited number of patients included in the study, especially in the rifampicin-resistant group, may lead to type II errors, meaning that we fail to detect differences in the incidence of rifampicin resistance between time periods with statistical significance. On the other hand, to our knowledge, our study is the first to report exclusively on rifampicin resistance in PJIs over a period of 20 years. In addition, our study included 238 staphylococcal PJIs, the largest study population hitherto investigated on this specific topic. The retrospective study design means that limitations apply regarding inaccuracies or misinterpretations of information received from medical records. Most notably, we do not have information on which antibiotic was given in combination with rifampicin. However, since we adhere to the above-discussed guidelines that changed over time, we performed a sensitivity analysis for the different periods and did not detect clinically significant differences (Appendix A, Appendix A). The heterogeneity of the bacteria causing the PJIs in our study can also be considered a limitation. However, we performed a separate analysis on PJIs caused only by CoNS, which is the main causative agent with rifampicin resistance in PJIs, but did not find any differences in our results compared with the analyses performed on the entire cohort. Further sub-group analyses for different bacteria were not performed due to the small number of observations for each subtype, rendering risk estimates impossible to interpret due to the large estimation uncertainty. However, it is important to acknowledge that the severity of PJI varies depending on—amongst others—the presence of difficult-to-treat bacteria, such as the more aggressive subtypes, e.g., *Staph. lugdunensis*, or patient-related risk factors that we were unable to control for [41]. Another limitation is that rifampicin was not used in all cases where the bacteria were sensitive to the drug. On the other hand, we found that patients treated with rifampicin had better results than those who were not, and this finding strengthens our results on outcome differences between the PJIs caused by rifampicin-resistant and rifampicin-sensitive bacteria. The use of disc diffusion for antibiotic susceptibility testing could be considered a limitation, but the method is generally used for antibiotic susceptibility testing in Swedish clinical microbiology laboratories. When well-validated and standardised, the disc diffusion methodology yields results of equal quality to those from MIC determination at a lower cost and workload [42].

## 4. Materials and Methods

This is a retrospective single-centre cohort study on patients registered in a local PJI register at Uppsala University Hospital, Uppsala, Sweden, a tertiary PJI referral centre. Initially, all patients diagnosed with and surgically treated for PJI of a hip or knee replacement between 2001 and 2020 were identified (593 PJIs in 538 patients). Subsequently, medical charts were reviewed to identify cases for which susceptibility testing for rifampicin was performed for the bacteria causing the PJI; 238 staphylococcal PJIs in 238 patients were included in the study. All cases fulfilled the 2018 International Consensus Meeting (ICM) criteria [5] for PJI, and the causative PJI pathogen was identified by at least two cultures from periprosthetic synovial fluid or tissue samples.

Antibiotic susceptibility testing was performed using the disc diffusion method. The breakpoints for microbiology analyses used in the clinical routine were changed slightly during the study period. Until 2010, the breakpoints were used according to SRGA (the Swedish Reference Group for Antibiotics) [43], and for staphylococci and rifampicin, they were sensitive at >25 mm and resistant at <22 mm. From 2011, the breakpoints were adjusted according to NordicAST (Nordic Committee on Antimicrobial Susceptibility Testing) [44], and were sensitive at ≥26 mm and resistant at <23 mm for staphylococci and rifampicin. Screening for methicillin-resistant staphylococci was performed using cefoxitin. Suspected MRSA was further confirmed by the detection of the *mecA* gene.

Data on gender, age, joint (hip or knee), operated side, type of surgery prior to index surgery (primary, revision, DAIR, other), type of PJI revision surgery (DAIR, 1-stage, 2-stage), rifampicin resistance of the causative bacteria, and whether the patient was treated with rifampicin or not, as well as the length of treatment where applicable, were collected from patients’ medical charts.

Revision surgery for PJI was defined as either DAIR, one-, or two-stage revision surgery. Patients may have undergone more than one revision for PJI; therefore, we defined the chronologically last PJI revision for which susceptibility data for rifampicin were available as the index surgery. The resistance to rifampicin of the causative bacteria was assessed using the routine disc diffusion methodology in the clinical microbiology laboratory at Uppsala University Hospital according to the guidelines of the Nordic Committee on Antimicrobial Susceptibility Testing. Treatment failure was defined as the need for any further surgical procedure related to the index PJI (i.e., new DAIR, implant removal, or amputation), PJI-related death, or the need for long-term suppressive antimicrobial treatment because of clinical signs of persistent or relapsing PJI. Infection-free arthroplasties were defined as cases where none of the signs of treatment failure were noted in the charts until the end of the follow-up period of the study (16 February 2022). Patients who had any competing event, such as revision surgery for reasons other than persistence or recurrence of the PJI, death, or loss to follow-up, were censored in the statistical analysis.

The primary outcome of the study was the incidence of rifampicin-resistant bacteria causing PJI over time (2001–2020) assessed in 5-year intervals. The secondary outcome was infection-free survival after PJI revision caused by either rifampicin-resistant or rifampicin-sensitive bacteria.

### Statistics

Continuous data are presented as means, medians, and ranges. Estimation uncertainty was assessed by calculating 95% confidence intervals (CIs). Categorical data were cross-tabulated, and proportions were investigated using the chi-squared test. The prevalence of rifampicin resistance over the observation period is presented in 5-year intervals (2001–2005, 2006–2010, 2011–2015, and 2016–2020). Kaplan–Meier survival analysis was used for the estimation of cumulative survival free from treatment failure, defined as described earlier. Differences between groups were investigated using the Mantel–Haenszel log-rank test. Multivariable Cox regression models were fitted to calculate the adjusted hazard ratios (HRs) for the risk of relapse with CIs for patients exposed to rifampicin-resistant bacteria compared with those not exposed. The following covariates were adjusted for joint, gender, age at the time of the index operation (categorised into the four groups: 16–64, 65–74, 75–84, and ≥85 years), and the type of surgery prior to the index surgery, as well as the type of index surgery itself. Log–log plots and Schoenfeld residuals indicated that the proportional hazards assumption was fulfilled in all models. The level of statistical significance was set at *p* < 0.05 in all analyses. The analyses were conducted using SPSS (version 27.0, accessed on 9 August 2023) and R (version 4.2.2, accessed on 9 August 2023) software.

## 5. Conclusions

The incidence of PJI caused by rifampicin-resistant bacteria has not increased over the past two decades in our study population. This is a reassuring observation since rifampicin-resistant bacteria present with a more than four-fold increased risk of treatment failure compared with PJI caused by rifampicin-sensitive bacteria, which highlights the importance of limiting rifampicin resistance. PJI caused by rifampicin-resistant bacteria warrants consideration of a different and maybe more aggressive treatment strategy in those difficult-to-treat PJIs. Future research on the incidence of the rifampicin resistance in PJI—even for different sub-groups, as determined by pathogens—is essential for the development of effective clinical management.

## Figures and Tables

**Figure 1 antibiotics-12-01499-f001:**
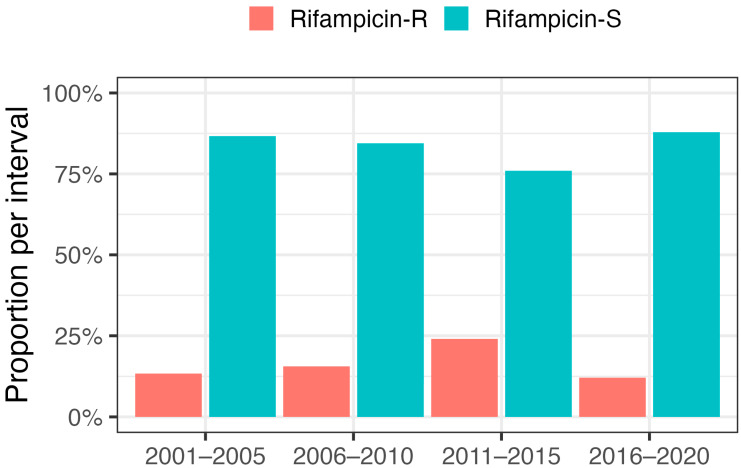
Proportion of bacteria sensitive (S) or resistant (R) to rifampicin over time, presented in 5-year periods.

**Figure 2 antibiotics-12-01499-f002:**
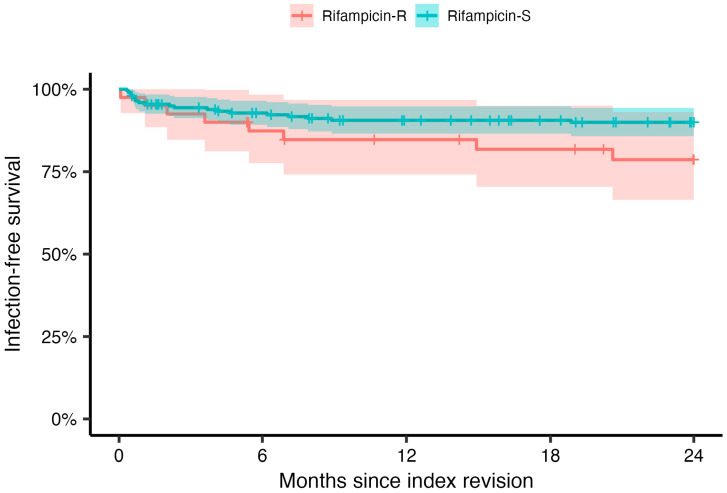
Kaplan–Meier analysis for 2-year infection-free survival over time for patients exposed to rifampicin-resistant (red) or -sensitive (blue) bacteria, regardless of underlying pathogen. The shaded areas indicate 95% confidence intervals and the vertical ticks censoring.

**Figure 3 antibiotics-12-01499-f003:**
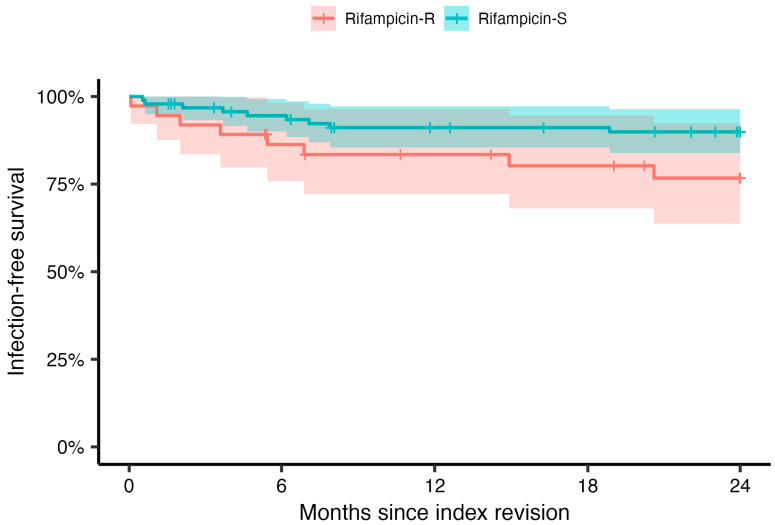
Kaplan–Meier analysis for 2-year infection-free survival over time for patients exposed to rifampicin-resistant (red) or -sensitive (blue) coagulase-negative staphylococci. The shaded areas indicate 95% confidence intervals and the vertical ticks censoring.

**Table 1 antibiotics-12-01499-t001:** Cohort characteristics presented by rifampicin susceptibility.

	Total Cohort (*n* = 238), *n* (%)	Rifampicin-Sensitive (*n* = 198), *n* (%, 95%CI)	Rifampicin-Resistant (*n* = 40), *n* (%, 95%CI)
Sex			
Male	137 (58)	118 (60, 52–66)	19 (48, 32–64)
Female	101 (42)	80 (40, 34–48)	21 (52, 36–68)
Joint			
Hip	154 (65)	129 (65, 58–72)	25 (63, 46–77)
Knee	84 (35)	69 (35, 28–42)	15 (38, 23–54)
Age			
16–64 years	67 (28)	58 (29, 23–36)	9 (23, 11–39)
65–74 years	76 (32)	59 (30, 24–37)	17 (43, 27–59)
75–84 years	62 (26)	50 (25, 19–32)	12 (30, 17–47)
≥85 years	33 (14)	31 (16, 11–22)	2 (5, 1–18)
Surgery prior to index surgery			
Primary arthroplasty	149 (63)	134 (68, 61–74)	15 (38, 23–54)
Revision arthroplasty	34 (14)	29 (15, 10–21)	5 (13, 5–28)
DAIR ^1^	40 (17)	24 (12, 8–18)	16 (40, 25–57)
Other	15 (6)	11 (6, 3–10)	4 (10, 3–25)
Index surgery			
DAIR ^1^	159 (67)	144 (73, 66–79)	15 (38, 23–54)
2-stage	63 (26)	40 (20, 15–27)	23 (58, 41–73)
1-stage	16 (7)	14 (7, 4–12)	2 (5, 1–18)
Causative bacteria			
Staphylococcus aureus	107 (43)	104 (53, 45–60)	3 (7, 2–21)
Coagulase-negative staphylococci	131 (53)	94 (47, 40–55)	37 (93, 79–98)

^1^ Debridement, antibiotics, and implant retention. *n*: number; CI: confidence interval

## Data Availability

Study data cannot be shared due to national legislation that limits access to register data to authorised researchers only, and because the ethical permission allows us to only report aggregate data. However, after obtaining ethical consent, the original datasets can be obtained from the Human Research Ethics Committee in Uppsala, Sweden.

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
