# Peer review of "Incidence of Rifampicin Resistance in Periprosthetic Joint Infection: A Single-Centre Cohort Study on 238 Patients"

_antibiotics, 2023, doi:10.3390/antibiotics12101499_

Round 1

Reviewer 1 Report

Thank you for thanking you for giving me the opportunity to review this manuscript This is an interesting paper that present how resistance to a key antibiotic in PJI, such as rifampicin has changed in a specific cohort (248 patients in  Sweden) over a long period of time (20 years).

The topic is really interesting, and the paper is well written, clear, and suitable for the reader and add a new information that from my knowledge no previous paper showed.

IN my opinion, some points have to be clarified to improve the value of this manuscript for readers. I suggest some changes:

The authors included 248 PJI, 238 due to staph. family pathogen and 10 mores cases with other gram +. From my pot of view, these 10 cases don’t add any extravalue to the paper (no increase enough the number of patients, or didn´t help to get extra conclusion) and in other way, the add a confusion factor due to any reader can assume that the good result observed with rifamp in Staph PJI will be the same in other gran + PJI. So, I recommend to remove these 10 no Staph. cases. 

Success of rifampicin PJI together with the increase of resistances are two major concerns with the Rifamp. treatment of Staph. PJI. However, there is one more important factor that is its tolerance. Some repost showed that around 30% patients don’t tolerate rifampicin treatment due to itching, liver disorder or drug interactions. Some authors suggest decrease the doses of Rifampicin or increase the time between administration, however these “alternatives” don’t have been evaluated until today.  IN this paper, rifampicin group have better succeed that no rifampicin group (resistance group) and it would be of great value to readers if authors can include the dose of rifampicin that they used, and if the know what percent of patients need to adjust doses or change to other antibiotic due to intolerance. 

My last concern is that there are a significative higher number of DAIR in Rifampicin sensitive (72%) VS (40%) in rifampicin resistant, and the opposite with2 stage. (26 VS 55). The authors explain this point in their limitation (line 149-155) in tern that due to DAIR results usually are worse that 2 stage. However, this affirmation is only true in case of same type of infection. It could be than in rifamp resistant group there are more chronic infection, or more complex (more previous surgery, tumor implant, etc) cases that could explain worse success. I recommend to clarify this point adding information of type infection in each group or highlighting that the type of infection was the same in both groups. IN cases that the authors don’t have this information it could be discussed in the limitation section as a possible bias. 

Author Response

Kind regards ,

Stergios Lazarinis

Reviewer 2 Report

I value the opportunity to review this interesting manuscript. Incidence of Rifampicin Resistance in Periprosthetic Joint Infection: A Single-Center Cohort Study on 248 Patients” may appropriate for consideration of publication in the Antibiotic. However, there are some major concern about this manuscript.

Please arrange the keywords alphabetically for a standardized presentation.

The references cited in this manuscript seem appropriate and support the statements made. However, it may be beneficial to include a few more recent references to ensure the information is up-to-date. Notably, the manuscript has been cited in 50.00% (12/24) of current publications within the preceding five years, which include 2 articles in 2019, 2 articles in 2020, 1 articles in 2021, 7 articles in 2022, and 0 articles in 2023.

The introduction part needs to be improved.  Although the significance of periprosthetic joint infection (PJI) and the utilization of surgical intervention alongside antibiotic treatment is briefly acknowledged, there seems to be a slight deficiency in explicitly delineating the specific objectives of the study.

Given the epidemiological focus of the work, it is essential to draw attention to the fact that there isn't any actual data on the topic at hand in the introduction. To enrich the study's framework and offer a more thorough context, I advise incorporating more data on the prevalence of periprosthetic joint infection and rifampicin-resistant bacteria.

I kindly request the authors to provide additional details regarding the methods employed to identify methicillin-resistant bacteria. Specifically, it would be valuable to know if the authors used cefoxitin or oxacillin for confirmation. Including this information will enhance the clarity and reproducibility of the study methodology. 

Could you please provide more details regarding the breakpoint used to determine resistance to rifampicin? 

Could you discuss the limitations of using the disc diffusion method for assessing rifampicin resistance compared to the the Minimum Inhibitory Concentration (MIC)? Understanding the implications of using the disc diffusion method in this study would be valuable for assessing the accuracy and reliability of the resistance assessment.

Given the limited number of MRSA isolates identified in the study, it might be challenging to conduct robust statistical analyses. Considering that only one MRSA isolate was identified in the study, it may not be necessary to separate it from Staphylococcus aureus.

Regarding the absence of Methicillin-resistant coagulase-negative staphylococci (MRCoNS) in the study findings, it is surprising and warrants further discussion. MRCoNS are known to be a significant cause of healthcare-associated infections. Their absence in the study raises questions about the methodology, sample selection, or potential limitations.

I suggest the authors include additional 95% confidence intervals (CI) in table 1 and 2 to better understand the data's variability and precision. This will strengthen the statistical robustness and reliability of the study. It would be beneficial to discuss the rationale and feasibility of calculating and presenting these additional CIs where applicable.

Please consider including additional information on the pharmacokinetic and pharmacodynamic effects of rifampin, if available, in the discussion section. Examining these effects can provide valuable insights into the observed disparities in epidemiological outcomes, enhancing the understanding of the study findings.

The conclusion section of this manuscript is quite vague and lengthy. The primary revelation from the significant content that was acquired should be stated in some conclusions, along with suggestions for clinical application. Furthermore, it is recommended to include a section in the conclusion that outlines potential directions for future research related to the current study.

Lastly, I would like to express my gratitude for the opportunity to review this submission. While the topic is indeed valuable and interesting, it is essential to provide additional interpretation and motivation to further enhance the manuscript.

Author Response

Kind regards,

Stergios Lazarinis

Reviewer 3 Report

The authors have conducted a time trend analysis of rifampicin resistance in periprosthetic joint infections. Few points that need to be addressed are as mentioned below:

1.     In Introduction, a note on any latest guidelines on surgical and antibiotic therapy for treatment of PJI may be added.

2.     Which were the other antibiotics given in the hospital in combination with rifampicin for PJIs? Were there any change in the treatment protocols through the 20 year study period especially with respect to concomitant antibiotic therapy or other infection, prevention and control (IPC) practices which may confound the outcomes?

3.     Few more studies may be cited and their results compared with the findings of current study. For example,

·       Achermann Y, Eigenmann K, Ledergerber B, Derksen L, Rafeiner P, Clauss M, Nüesch R, Zellweger C, Vogt M, Zimmerli W. Factors associated with rifampin resistance in staphylococcal periprosthetic joint infections (PJI): a matched case-control study. Infection. 2013 Apr;41(2):431-7. doi: 10.1007/s15010-012-0325-7.

·       Krizsán G, Sallai I, Veres DS, Prinz G, Kovács M, Skaliczki G. Investigation of the effect of rifampicin resistance and risk factors on recovery rates after DAIR procedure in patients with prosthetic joint infection. J Orthop Surg Res. 2023 Aug 21;18(1):611. doi: 10.1186/s13018-023-04091-y.

4.     Did the authors observe any difference in the incidence of rifampicin resistance and /or outcomes between immediate and delayed administration of the drug? Please mention and compare with published literature.

·       Darwich A, Dally FJ, Bdeir M, Kehr K, Miethke T, Hetjens S, Gravius S, Assaf E, Mohs E. Delayed Rifampin Administration in the Antibiotic Treatment of Periprosthetic Joint Infections Significantly Reduces the Emergence of Rifampin Resistance. Antibiotics (Basel). 2021 Sep 21;10(9):1139. doi: 10.3390/antibiotics10091139.

5.     The characteristics of PJIs caused by different types of staphylococci may be discussed.

·       Lourtet-Hascoët J, Bicart-See A, Félicé MP, Giordano G, Bonnet E. Staphylococcus lugdunensis, a serious pathogen in periprosthetic joint infections: comparison to Staphylococcus aureus and Staphylococcus epidermidis. Int J Infect Dis. 2016 Oct;51:56-61. doi: 10.1016/j.ijid.2016.08.007).

Minor editings required.

Author Response

Please the attachment

Kind regards,

Stergios Lazarinis

Round 2

Reviewer 2 Report

Based on these revisions, it is evident that you have taken my feedback seriously and made substantial improvements to the manuscript. I am impressed with the significant efforts you have made to address the raised concerns. You have not only provided additional information and relevant references but have also improved the clarity and reproducibility of the study methodology. I would like to express my appreciation for the discussion on the study limitations and the explanations for the absence of certain findings.